# Association between Early Phase Serum Albumin Levels and Outcomes of Post-Cardiac Arrest Patients: A Systematic Review and Meta-Analysis

**DOI:** 10.3390/jpm12111787

**Published:** 2022-10-29

**Authors:** Heekyung Lee, Juncheol Lee, Hyungoo Shin, Tae-Ho Lim, Bo-Hyoung Jang, Youngsuk Cho, Wonhee Kim, Jae-Guk Kim, Kyu-Sun Choi, Min-Kyun Na, Chiwon Ahn, Sae-Min Kwon

**Affiliations:** 1Department of Emergency Medicine, Hanyang University College of Medicine, Seoul 04763, Korea; 2Department of Preventive Medicine, College of Korean Medicine, Kyung Hee University, Seoul 02447, Korea; 3Department of Emergency Medicine, College of Medicine, Hallym University, Chuncheon 24253, Korea; 4Department of Neurosurgery, Hanyang University College of Medicine, Seoul 04763, Korea; 5Department of Emergency Medicine, College of Medicine, Chung-Ang University, Seoul 06974, Korea; 6Department of Neurosurgery, Dongsan Medical Center, Keimyung University School of Medicine, Daegu 42601, Korea

**Keywords:** albumin, heart arrest, patient outcome assessment, meta-analysis

## Abstract

We aimed to evaluate early phase serum albumin levels in and outcomes of resuscitated patients after cardiac arrest. Medline, EMBASE, and the Cochrane Library were systematically searched until 4 July 2022, for studies on post-cardiac arrest patients and involving measurement of early phase albumin levels and assessment of in-hospital mortality or neurologic outcomes. Two reviewers independently assessed the methodological quality of the included studies using the Quality in Prognosis Studies tool. We included 3837 patients from seven observational studies in this systematic review and meta-analysis. The serum albumin level was significantly higher in survivors than in non-survivors, showing a positive association with an overall standardized mean difference (SMD) [(mean value of non-survivors—mean value of survivors)/pooled standard deviation] of 0.55 (95% confidence interval [CI], 0.48–0.62; I^2^ = 0%; *p* < 0.001). Additionally, the serum albumin level was significantly higher in the good neurologic outcome group than in the poor neurologic outcome group (four studies; SMD = 1.01, 95% CI = 0.49–1.52, I^2^ = 87%; *p* < 0.001). Relatively low serum albumin levels in the early phase may be associated with in-hospital mortality of resuscitated patients after cardiac arrest. However, we could not evaluate the association between albumin level and neurologic outcome because of limited included studies and unresolved high heterogeneity.

## 1. Introduction

Ischemia-reperfusion injury that occurs during cardiac arrest (CA) and return of spontaneous circulation (ROSC) after cardiopulmonary resuscitation (CPR) is referred to as post-cardiac arrest syndrome (PCAS) [1]. Hypoxic-ischemic cerebral injury is the major cause of mortality and permanent neurological injury in survivors of both out-of-hospital cardiac arrest (OHCA) and in-hospital cardiac arrest (IHCA) [2,3]. Accurate and early prognosis is essential to prevent unnecessary treatment when a poor outcome is inevitable and to avoid inappropriate cessation of aggressive treatment for patients who might have the potential to recover from PCAS [4]. However, there is no gold standard tool to accurately predict outcomes, and the current guidelines recommend multimodality assessment for neuroprognostication [5].

Serum biomarkers as outcome predictors of patients with PCAS have several advantages, including relatively easy access, rapid performance, immediate results, and repeatability [5]. Neuron-specific enolase (NSE) and S100 calcium-binding protein B (S100B) are the most widely studied and are recommended in recent guidelines as prognostic biomarkers [5]. However, early phase use of NSE as a predictor is not useful, because a minimum of 24 h after ROSC is required for using NSE concentration to predict outcomes [6,7,8]. Albumin plays several important physiological roles such as scavenging free radicals by maintaining plasma oncotic pressure, antioxidant properties, microvascular integrity, and anti-inflammatory activity [9]. Because of these roles, previous studies have reported that the serum albumin level is a predictor of outcomes of critically ill patients with conditions such as sepsis and myocardial infarction [10,11,12]. Serum albumin levels might also be a predictor of outcomes in the early phase of CA. Several studies have reported that low serum albumin levels are associated with poor outcomes of CA patients [13,14,15]. However, the previous studies had limitations in that their conclusions could not be generalized due to a limited sample size and heterogeneity of assessment methods.

Hence, we performed the first systematic review and meta-analysis of previous studies to assess the association between serum albumin levels and outcomes of patients hospitalized after CA. We hypothesized that in resuscitated patients, there would be an association between early phase low serum albumin levels and in-hospital mortality and poor neurologic outcomes (PNO).

## 2. Materials and Methods

### 2.1. Protocol and Registration

This study was performed according to the principles outlined in the Meta-analysis of Observational Studies in Epidemiology (MOOSE) and Preferred Reporting Items for Systematic Reviews and Meta-analysis (PRISMA) guidelines [16,17]. We registered the protocol at PROSPERO (http://www.crd.york.ac.uk/PROSPERO/, accessed on 20 December 2021), with registration number CRD42022299361.

### 2.2. Eligibility Criteria

A literature search of critical assessments was performed, and eligible studies were selected and evaluated through a systematic review and meta-analysis. The PICO (population, intervention, comparison, and outcome) question was as follows: population (P) = adult patients with ROSC after CA; intervention (I) = serum albumin level; comparator (C) = none; and outcome (O) = in-hospital mortality and neurologic outcome.

### 2.3. Information Sources and Search Strategy

An extensive database search was performed for studies that assessed the prognostic impact of serum albumin levels in adult patients with ROSC after CA. Two reviewers (H. Lee and J. Lee) performed the literature search on 4 July 2022, using the EMBASE (from 1974 to 1 July 2022) and MEDLINE (from 1946 to 1 July 2022) databases through the Ovid interface and the Cochrane Library (all years). The search terms included “heart arrest” or “cardiac arrest” or “cardiopulmonary resuscitation” or “return of spontaneous circulation” and “albumin” (Appendix A). All articles that reported any prospective or retrospective cohort study that addressed our PICO question were included in this meta-analysis, and no language restrictions were used. A manual search was performed, and references of eligible studies were checked to identify additional relevant studies.

### 2.4. Study Selection

Two experienced reviewers (H. Lee and J. Lee) independently screened the titles and abstracts of all selected studies, and irrelevant studies were excluded. A full-text review of potentially relevant articles was subsequently performed. The exclusion criteria were as follows: irrelevant outcomes (main outcome not eligible for this meta-analysis), irrelevant populations (patients who did not experience sustained ROSC), duplicate data or studies, animal studies, reviews, case reports, and conference abstracts. Any disagreement not resolved on discussion was reviewed by a third reviewer (H. Shin), and differences were resolved through discussion until a consensus was reached.

### 2.5. Data Collection Process and Data Items

Two reviewers (H. Lee and J. Lee) independently extracted relevant data that included study characteristics, main outcomes, and basic characteristics of patients in the included studies. The following study variables were extracted: name of the first author, publication year, country of study, inclusion period, study type, CA type (OHCA or IHCA), sample size, number of patients with in-hospital mortality and PNO, serum albumin measurement time, and time points of outcome measurement. The neurological outcome was classified as good or poor based on the cerebral performance category (CPC; 1–2 = good neurologic outcome [GNO]; 3–5 = PNO) scores (CPC1, good cerebral performance; CPC2, moderate cerebral disability; CPC3, severe cerebral disability; CPC4, coma or vegetative state; and CPC5, brain death or death). The mean (±SD) serum albumin levels of the patients were calculated from the median values with interquartile ranges [18].

### 2.6. Study Risk of Bias Assessment

Two reviewers (H. Lee and J. Lee) independently assessed the methodological quality of the included studies, with the authorship and journal data blinded. Unresolved disagreements between the reviewers were resolved through a discussion. Six bias domains (study participation, study attrition, prognostic factor measurement, outcome measurement, study confounding, and statistical analysis and reporting) were assessed using the Quality in Prognosis Studies (QUIPS) tool in systematic reviews [19]. The QUIPS tool was used, with values of 0, 1, and 2 considered to indicate high, unclear, and low risk, respectively [19]. Studies that achieved ≥9 points in the sum of each domain score were defined as high-quality, and <9 points were defined as low-quality.

### 2.7. Synthesis Methods

In the main analysis, we identified the prognostic impact of serum albumin levels in predicting in-hospital mortality and PNO in patients with ROSC after CA. The strength of the association between decreased serum albumin levels and in-hospital mortality and PNO was estimated with standardized mean differences (SMD) using a random-effects model [20]. This model was also used to estimate individual data of the included studies, considering the diversity of countries, inclusion period, CA type, and serum albumin measurement time. The serum albumin level across comparisons between non-survivors and survivors, PNO, and GNO were presented as mean differences with 95% confidence intervals (CIs). I^2^ statistics were used to quantify heterogeneity and estimate the proportion of inter-study inconsistency resulting from the true differences between studies, with values of 25%, 50%, and 75% categorized as low, moderate, and high, respectively [21].

We conducted additional analyses, including sensitivity and meta-regression analyses, if the heterogeneity was moderate or high in the main meta-analysis. To interpret the potential causes of inter-study heterogeneity, a sensitivity analysis was conducted by sequentially omitting the studies. Meta-regression analysis was conducted to identify heterogeneity and assess the influence of study characteristics on the results. Funnel plots were used to assess publication bias. The asymmetry of the funnel plot indicates the presence of bias.

The reference management software Endnote X9 (Clarivate Analytics, Philadelphia, PA, USA) was used to organize all studies identified in the literature search. Statistical analysis was performed using Review Manager (version 5.3; Cochrane Collaboration, Oxford, UK). Identification of publication bias, sensitivity analysis, and meta-regression were performed using the R packages “meta” and “metafor” (R version 4.0.4; R Foundation for Statistical Computing, Vienna, Austria). Statistical significance was set at *p* < 0.05.

### 2.8. Certainty Assessment

The Grading of Recommendations, Assessment, Development, and Evaluation (GRADE) framework for meta-analysis was used to evaluate the quality of evidence, and the level of evidence was graded as high, moderate, low, or very low [22]. The level of evidence was determined using seven domains (participants, design, risk of bias, inconsistency, indirectness, imprecision, and publication bias), and the importance of the outcome was classified as critical, important, or not important [22]. It was conducted using GRADEprofiler (version 3.6.1, The GRADE Working Group), and a summary of findings was presented with an evidence profile.

## 3. Results

### 3.1. Study Selection

A total of 370 studies were identified through a database search along with two additional manual searches (Figure 1); 43 duplicate records were removed, and 315 were excluded after assessing both titles and abstracts. After the full texts of the 12 remaining articles were reviewed, we excluded five articles for the following reasons: irrelevant outcomes (n = 4) and duplicate data (n = 1). Finally, seven observational studies that enrolled total 3837 patients were included in the meta-analysis [13,14,15,23,24,25,26]. The reasons why potentially relevant studies failed to meet the eligibility criteria are presented in Appendix A.

### 3.2. Study Characteristics

The main characteristics of the included studies are summarized in Table 1. The baseline characteristics of the enrolled patients are presented in Appendix A. Seven observational studies were published between 2017 and 2022 [13,14,15,23,24,25,26]. Two single-center studies were conducted in Turkey [23,24]. One multicenter study and three single-center studies were conducted in Korea [13,15,25,26], while one multicenter study was conducted in China [14]. Five studies included only OHCA patients [13,23,24,25,26]. Six studies reported in-hospital mortality [13,14,23,24,25,26], and four studies reported neurologic outcomes as the outcome of patients [13,15,25,26]. The mortality rate was 59.0% in six studies. The poor neurologic outcome rate was 81.0% in four studies.

### 3.3. Risk of Bias in Studies

The results of the quality assessment of the included observational studies are presented in Appendix A, and the detailed rating of reports is presented in Appendix A. All studies showed a low risk of bias in the following four domains: study participation, prognostic factor measurement, outcome measurement, statistical analysis and reporting. Four studies showed a high risk of bias in the domain of study attrition [13,23,24,25], and three studies showed a high risk of bias in the domain of study confounding [14,23,26]. Among the seven included studies in this meta-analysis, per the quality scoring system, one study had low quality [23].

### 3.4. Results of Meta-Analyses

#### 3.4.1. Value of the Serum Albumin Level for Predicting In-Hospital Mortality

The serum albumin level was found to be relatively higher in survivors than in non-survivors, showing a positive association with an overall SMD ([mean value in the non-survivors − mean value in survivors]/pooled SD) of 0.55 (95% CI, 0.48–0.62; I^2^  =  0%; *p*  <  0.001, Figure 2A).

A meta-analysis of in-hospital mortality was performed separately for studies that included only OHCA patients, and this analysis showed a significant difference between the two groups (*p* < 0.001, Figure 3A).

#### 3.4.2. Value of the Serum Albumin Level for Predicting Poor Neurological Outcome

The serum albumin level was significantly higher in the GNO group than in the PNO group (four studies; SMD = 1.01, 95% CI = 0.49–1.52, I^2^  =  87%; *p*  <  0.001, Figure 2B).

A meta-analysis of neurologic outcome was performed separately for studies that included only OHCA patients, and this analysis showed a significant difference between the two groups (*p* < 0.001, Figure 3B).

#### 3.4.3. Value of the Serum Lactate Level for Predicting In-Hospital Mortality

The serum lactate level was found to be significantly higher in non-survivors than in survivors, showing a negative association with an overall SMD ([mean value in the survivors − mean value in non-survivors]/pooled SD) of 0.54 (95% CI, 0.27–0.82; I^2^  =  87%; *p*  <  0.001, Figure 4A).

#### 3.4.4. Value of the Serum Lactate Level for Predicting Poor Neurological Outcome

The serum lactate level did not show significant difference between the GNO and PNO groups (two studies; SMD = 0.71, 95% CI = −0.21–1.62, I^2^  =  86%; *p*  =  0.13, Figure 4B)

### 3.5. Additional Analysis for Identifying and Measuring Heterogeneity

Subgroup analysis could not be performed because of the lack of included studies. A summary of the sensitivity analysis of serum albumin level for PNO is presented in Appendix A. However, no significant decrease in heterogeneity (less than moderate heterogeneity) was found after omitting each study. Moreover, there was no significant influence of five characteristics (sample size, male sex, initial shockable rhythm, witness arrest, and proportion of targeted temperature management on heterogeneity according to the meta-regression analysis (Appendix A).

### 3.6. Risk of Bias across Studies

There was no obvious asymmetry between the forest plots. The contour-enhanced funnel plot, which was used to evaluate reporting bias such as publication bias, revealed no significant asymmetry (Appendix A).

### 3.7. Certainty of Evidence

The results of this study were assigned a low level of evidence according to the evidence profile, using the GRADE framework (Appendix A). The main reason for this was the analysis of observational studies. The importance of the result was determined to be critical because the prediction of survival and neurologic outcomes in post-CA is crucial for deciding the management plan.

## 4. Discussion

In the present study, we aimed to investigate the association between early phase serum albumin levels, survival, and neurologic outcomes in post-CA patients via a systematic review and meta-analysis. We found that patients who survived CA had higher serum albumin levels than non-survivors in the early phase of ROSC. To the best of our knowledge, this is the first meta-analysis to investigate the association between serum albumin levels and outcomes in post-CA patients.

Serum albumin, which is the main protein in plasma, is a determinant of plasma oncotic pressure and modulates fluid distribution in body compartments [27]. Serum albumin has been widely used as a valuable marker and therapeutic agent in critically ill patients, including those with shock, hypovolemia, cancer, acute respiratory distress syndrome, liver disease, resuscitation, and PCAS [12,28,29,30]. Hypoalbuminemia can be caused by low energy supply, increased loss of serum albumin, decreased liver synthesis, increased catabolism, and distribution failure of serum albumin [31]. In particular, increased vascular permeability and transcapillary loss of albumin are major causes of acute hypoalbuminemia in systemic inflammatory states, including sepsis and PCAS [32]. Several animal studies have proposed the protective effects of human serum albumin, including anti-inflammatory effects and reduction of ischemia-reperfusion injury [33,34]. Furthermore, albumin synthesis in the liver is significantly reduced in the inflammatory state [35]. Additionally, serum albumin level is a biochemical marker of nutritional status and is strongly associated with malnutrition severity, which is regarded as a general risk factor in critically ill patients. For example, Takegawa et al. presented that the association between hypoalbuminemia and poor prognosis in critically ill patients is independently affected by nutritional status and inflammation [36].

Reliable prognostic indicators are crucial for establishing a management plan for patients with ROSC. Despite advances in resuscitation research, accurate prediction of survival outcomes after ROSC remains a challenge [37]. The current guidelines recommend the use of NSE, in combination with other prognostic tools, as a biomarker for neuroprognostication. However, NSE should not be collected for a minimum of 24 h after cardiac arrest, and is, therefore, not suitable as an early phase predictor of post-CA outcomes [6,7,8].

Alternatively, decreased serum albumin levels are associated with mortality and can be used as a prognostic marker in critically ill patients [38]. Several previous studies have investigated albumin concentration as an early phase predictor of outcomes in patients with CA. Kong et al. reported that the initial albumin concentration was significantly higher in the survival group than in the non-survival group and in the GNO group than in the PNO group among OHCA survivors [39]. Another retrospective study revealed that albumin levels within 10 min of the ED visit was higher in the survival group than in the non-survival group with an area under the ROC curve of 0.72 (95% CI, 0.66–0.78) [24]. In the present meta-analysis, the relationship between serum albumin levels in the early phase of CA and either survival or neurologic outcomes was analyzed in the seven included studies. The SMD of the serum albumin level was significantly lower in the non-survival and PNO groups, with heterogeneity of I^2^ = 0% and I^2^ = 87%, respectively. However, the number of studies included in each analysis was relatively small, and the heterogeneity of the neurologic outcome analysis was high. We were unable to interpret the factors influencing heterogeneity despite additional analyses including sensitivity and meta-regression analysis. However, the SMD of serum albumin level was significantly lower in the survival group than in the non-survival group, with low heterogeneity (I^2^ = 0%).

In critically ill patients, blood lactate level has been commonly used as an indication of tissue perfusion changes for decades [40,41]. Hyperlactatemia, caused by systemic ischemia, is associated with a high mortality rate after both IHCA and OHCA, as well as other severe illnesses, including sepsis [42]. A previous meta-analysis reported that lactate concentration on admission was related to neurologic outcomes after CA [43]. In the present study, we analyzed the relationship between lactate level and mortality or neurologic outcomes from five of the seven included studies that reported both albumin and lactate levels. We attempted to compare lactate levels with albumin levels for understanding their predictive performance of outcomes of post-CA patients. However, owing to the unresolved high heterogeneity and small number of included studies, we could not draw conclusions about the association between lactate levels and in-hospital mortality or neurologic outcome. The high heterogeneity and inconsistent results compared with those of previous meta-analyses may have originated from selection bias and the small number of included studies. Interestingly, two recent studies reported a better prognostic value of the lactate/albumin ratio (LAR) than the sole use of lactate or albumin of patients with OHCA [24,39]. Meta-analysis of LAR was impossible because of the small number and irrelevant outcomes of reported studies.

Except for the relationship between albumin levels and in-hospital mortality, high heterogeneity was observed in this meta-analysis. The differences in study characteristics (including country, study type, location of CA (OHCA or IHCA), serum measurement time), and small number of included studies might be the reason for the high heterogeneity. One multicenter observational study that reported on albumin level and outcomes of OHCA patients [39] was excluded because duplicated data were present of another included study [26], and it is summarized in the supplemental materials.

This study had some limitations. First, there are many variables that affect the prognosis of cardiac arrest patients. However, meta-analysis cannot adjust for these variables because different studies have assigned somewhat different meanings to many of the variables. In addition, as stated previously, some statistical and clinical heterogeneities between the studies were not resolved. Despite this, this study was conducted using SMD, a random-effects model that accounts for the variation between studies to reduce heterogeneity. This could have limited the quality and interpretability of the findings. This limitation requires additional analyses such as subgroup analysis or meta-regression. However, this was limited by the lack of included studies. Second, we could not determine diagnostic test accuracy for calculating sensitivity, specificity, odds ratio, or summary ROC curve owing to lack of data. Third, the applicability of our findings is limited because the included research was conducted in three nations and over half of the included studies were conducted in Korea. Fourth, we did not consider confounding variables that could have affected albumin or lactate concentrations, such as underlying diseases, CA duration, nutritional status, drug or alcohol use, and smoking history. To improve the findings of this meta-analysis, a large-scale prospective follow-up investigation with a larger sample size is required. Finally, observational or selection bias could have occurred because we included observational studies in this meta-analysis, not randomized controlled trials. The result of GRADE was rated as “low” in present study. To overcome this limitation, serum albumin level should be evaluated for its predictive value for CA patients by well-designed randomized clinical trials.

## 5. Conclusions

Serum albumin levels measured in the early phase of post-CA were significantly higher in the survival group than in the non-survival group. However, the present study could not evaluate the association between albumin level and neurologic outcome because of the lack of included studies and unresolved high heterogeneity. The albumin concentration test could be helpful in predicting in-hospital mortality of post-CA patients. Owing to the low level of evidence provided by GRADE, a well-designed, large-scale, prospective trial is required to validate the findings of this study.

## Figures and Tables

**Figure 1 jpm-12-01787-f001:**
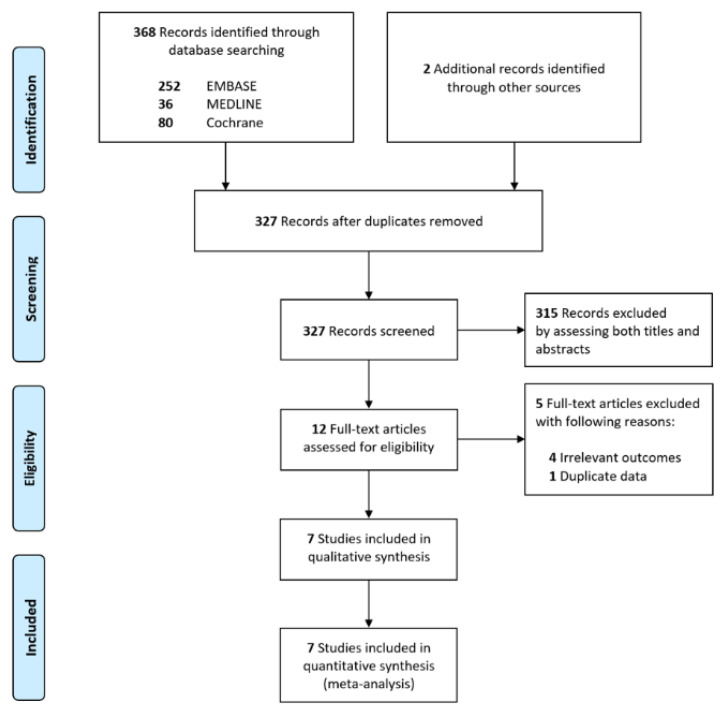
Flow diagram for the identification of relevant studies.

**Figure 2 jpm-12-01787-f002:**
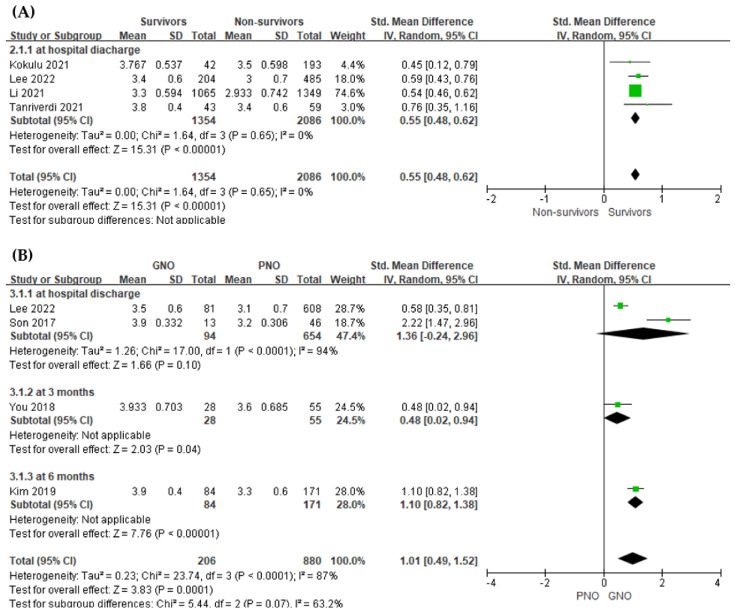
Forest plot of the association between the early phase serum albumin level and outcomes of post-cardiac arrest patients. (**A**) in-hospital mortality (**B**) neurologic outcome.

**Figure 3 jpm-12-01787-f003:**
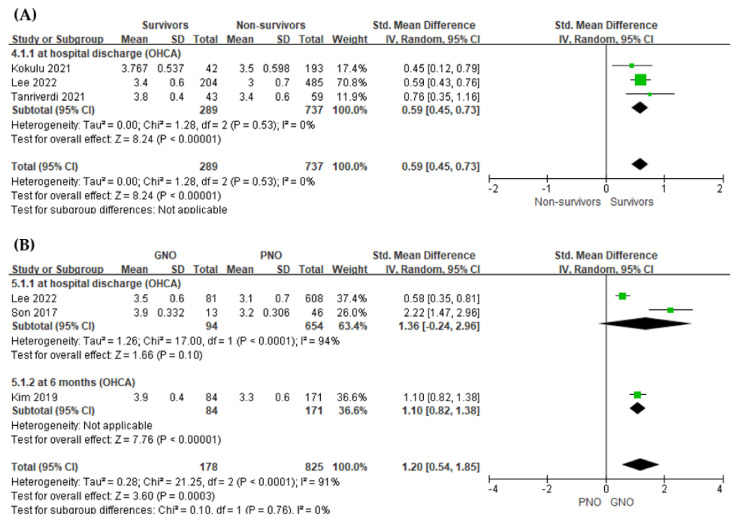
Forest plot of the association between early phase serum albumin level and outcomes of out-of-hospital cardiac arrest patients. (**A**) in-hospital mortality (**B**) neurologic outcome.

**Figure 4 jpm-12-01787-f004:**
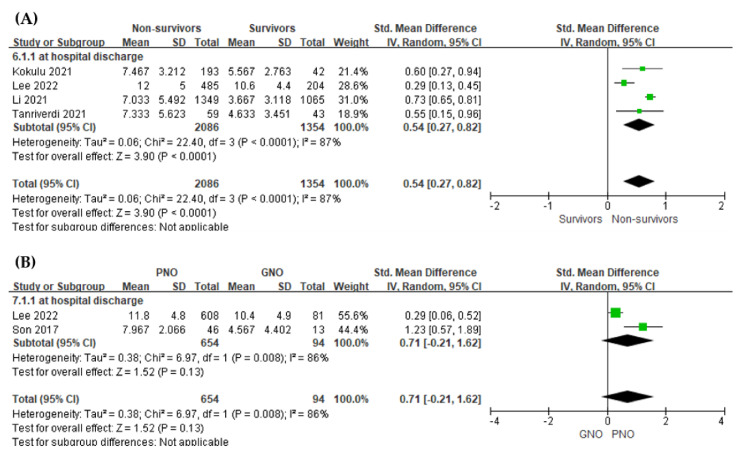
Forest plot of the association between early phase serum lactate level and outcomes of post-cardiac arrest patients. (**A**) in-hospital mortality (**B**) neurologic outcome.

**Table 1 jpm-12-01787-t001:** Study characteristics.

Author (Year)	Country	Inclusion Period	Study Type	Sample Size, n	In-Hospital Mortality, n (%)	PNO *, n (%)	Serum Sampling Time	Outcome Measurement Timepoint
Kim (2019)	Korea	2009–2016	sPOS	255 OHCA	107 (42.0)	171 (67.1)	within 1 h after ROSC	at 6 months
Kokulu (2021)	Turkey	January 2018–December 2020	sPOS	235 OHCA	193 (82.1)	NR	within 10 min on admission	at hospital discharge
Lee (2022)	Korea	October 2015–June 2020	mPOS	689 OHCA	485 (70.4)	608 (88.2)	immediately after ROSC	at hospital discharge
Li (2021)	China	2014–2015	mROS	2414 CA	1349 (55.9)	NR	within 24 h on admission	at hospital discharge
Son (2017)	Korea	March 2013–December 2015	sROS	59 OHCA	21 (35.6)	46 (78.0)	within 24 h after ROSC	at hospital discharge
Tanriverdi (2021)	Turkey	NR	sROS	102 OHCA	59 (57.8)	NR	immediately on admission	at hospital discharge
You (2018)	Korea	January 2014–January 2018	sROS	83 CA	NR	55 (66.3)	immediately on admission	at 3 months

Abbreviations: PNO = poor neurologic outcome; sPOS = single-center prospective observational study; OHCA = out-of-hospital cardiac arrest; ROSC = return of spontaneous circulation; NR = not reported; mPOS = multi-center prospective observational study; mROS = multi-center retrospective observational study; CA = cardiac arrest; h = hours; sROS = single-center retrospective observational study. * PNO was defined as cerebral performance categories 3–5.

## Data Availability

Not applicable.

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
