# Peer review of "Association between Early Phase Serum Albumin Levels and Outcomes of Post-Cardiac Arrest Patients: A Systematic Review and Meta-Analysis"

_jpm, 2022, doi:10.3390/jpm12111787_

Round 1

Reviewer 1 Report

The authors presented a meta-analysis studying the relation between the serum albumin levels in early phase of resuscitated patients after cardiac arrest and the in-hospital mortality and neurologic outcomes.

7 studies with 3837 patients were enrolled.

Serum albumin level was significantly higher in survivors and was associated in the good neurologic outcome.

The paper is well written. Statistical analysis was properly conducted.

I recommend its publication with no further revisions.

Author Response

To: Reviewer #1

Thank you for your detailed and rigorous comments. We further revised our manuscript based on all of your comments as follows. Sincerely.

Best regards.

Reviewer 2 Report

This interesting work is about the relationship between serum albumin level and associated outcomes in the patients with CA. The authors also found positive relationship between albumin level and neurological outcomes. All enrolled studies had the similar outcomes about high serum albumin predict good outcomes due to better nutritional level. The majority of population was from “Li et al”. The author may increase the discussion about lactate and outcomes.

I recommend adding back supplemental figure to main manuscript to discuss the issue about lactate level and associated outcomes in the patients with OHCA. The authors may add the discussion about the poor perfusion status and serum lactate level.

I recommend adding one figure about the patients with OHCA only and excluded the studies without pure OHCA.

The author should focus the influence of serum albumin and nutritional status and the effect on neurological outcomes.

Author Response

Reviewer #2

To. Reviewer #2

Thank you for your detailed and rigorous comments. We further revised our manuscript based on all of your comments as follows. Sincerely.

Best regards.

Reviewer 3 Report

To:

Editorial Board

Journal of Personalized Medicine

Title: “Association between Early Phase Serum Albumin Levels and Outcomes of Post-Cardiac Arrest Patients: A Systematic Review and Meta-Analysis”

Dear Editor,

I read this paper and I think that:

-          Lack of randomized controlled trials is a limitation of this metanalysis. This reduces the impact of the results. Please discuss such a point.

-          Many variables should be encountered when dealing with the outcome of patients who experienced cardiac arrest. A meta-analysis could not overcome such limitations. This should be deeply discussed.

Author Response

To. Reviewer #3

Thank you for your detailed and rigorous comments. We further revised our manuscript based on all of your comments as follows. Sincerely.

Best regards.

Reviewer 4 Report

In this manuscript, the authors searched literature from three databases and performed a meta-analysis on the association between serum albumin levels with in-hospital mortality and poor neurological outcome. With seven included studies, the author found that low serum albumin levels may be associated with a high risk of in-hospital mortality after cardiac arrest, which may provide potential guidance in clinical treatment for resuscitated patients after cardiac arrest. Several points below can help to improve the quality of the manuscript.

1.       In the risk of bias of included studies, the assessment of the risk of bias relies on two independent reviewers. Is there any unbiased way to assess the bias of the studies? In this study, one out of seven included studies was defined as low quality. Please include how to define the low quality of the studies in this meta-analysis.

2.       In the analysis of serum albumin level in figure 2, why is one of the studies excluded from the data for at hospital discharge?

3.       Is there any gender difference in early phase serum albumin post cardiac arrest based on meta-analysis?

4.       All supplemental table in the main text is wrong and there is no table S6 in the supplement tables. Please correct it.

Author Response

Reviewer #4

Dear Reviewer:

Thank you for your comments. We agree with your opinion about our study and manuscript.

Your insightful comments have helped us correct the errors and enhance the quality of the manuscript.

Our point-by-point responses to your comments are given below.

Best regards.

Reviewer 5 Report

H lee and other presents a comprehensive systematic review of the prognostic role of albumin concentration in PCAS patients.

The overall work is generally well written, well structured, and follower PRISMA indication, so I have only minor considerations.

- Explicit better the neurologic outcome

- Explicit better how did you use the quality in prognostic studies tool

- rewrite and explicit better the risk of bias assessment

- some periods in the discussion appeared hard to read and I suggest you revise language

Best Regards

Author Response

Reviewer #5

Dear Reviewer:

Thank you for your comments. We agree with your opinion about our study and manuscript.

We have made several changes in the manuscript in accordance with your suggestions.

We acknowledge that your insightful comments have helped us correct the errors and enhance the quality of the manuscript.

Our point-by-point responses to your comments are given below.

Best regards.

Round 2

Reviewer 2 Report

In the revised manuscript, the authors clearly respond to the comments. The work concluded serum albumin levels measured in the early phase of post-CA were significantly higher in the survival group than in the non-survival group. However, the present study could not evaluate the association between albumin level and neurologic outcome. The albumin concentration test could be helpful in predicting in-hospital mortality of post-CA patients.

Reviewer 3 Report

authors included limitations of their analysis. Indeed, these limitations still reduce the strenght of the paper.

Reviewer 4 Report

The authors answered all of my comments.